# Driving Fatigue Detection Based on the Combination of Multi-Branch 3D-CNN and Attention Mechanism

**Wenbin Xiang [1], Xuncheng Wu [1],*, Chuanchang Li [1], Weiwei Zhang [1,2,3] and Feiyang Li [4]**

[1] School of Mechanical and Automotive Engineering, Shanghai University of Engineering Science, Shanghai 201620, China; jzdsm7813@163.com (W.X.); lichuanchang@126.com (C.L.); zwwsues@163.com (W.Z.)
[2] Shanghai Smart Vehicle Integration Innovation Center Co., Shanghai 201620, China
[3] School of Vehicle and Mobility, Tsinghua University, Beijing 100089, China
[4] School of Information Science, Beijing Language and Culture University, Beijing 100083, China; strangepony@163.com
* Correspondence: wuxunchengsues@163.com; Tel.: +86-131-2207-9003

**Featured Application: this study proposes a new fatigue driving detection model based on the combination of 3D convolution and attention mechanism, which can evaluate the driver's fatigue state, reduce the accident rate and ensure the driver's safety.**

**Abstract:** Fatigue driving is one of the main causes of traffic accidents today. In this study, a fatigue driving detection system based on a 3D convolutional neural network combined with a channel attention mechanism (Squeeze-and-Excitation module) is proposed. The model obtains information of multiple channels of grayscale, gradient and optical flow from the input frame. The temporal and spatial information contained in the feature map is extracted by three-dimensional convolution, after which the feature map is fed to the attention mechanism module to optimize the feature weights. EAR and MAR are used as fatigue analysis criteria and, finally, a full binary tree SVM classifier is used to output the four driving states. In addition, this study uses the frame aggregation strategy to solve the frame loss problem well and designs application software to record the driver's status in real time while protecting the driver's facial privacy and security. Compared with other classical fatigue driving detection methods, this method extracts features from temporal and spatial dimensions and optimizes the feature weights using the attention mechanism module, which significantly improves the fatigue detection performance. The experimental results show that 95% discriminative accuracy is achieved on the FDF dataset, which can be effectively applied to driving fatigue detection.

**Keywords:** driving fatigue detection; 3D convolution; channel attention mechanism; facial privacy protection

## 1. Introduction

According to statistics from the National Bureau of Statistics of China, the global car ownership in 2021 is about 1.05 billion, of which 80% are sedans, and it is growing at a rate of 2–5% every year [1]. As the number of cars continues to grow rapidly, traffic accidents also occur more and more frequently. More than 95% of traffic accidents are related to the driver's behavior [2] and the driver's fatigue distracted behavior will affect his operating behavior, thereby causing traffic accidents. Therefore, the research and development of the detection system of driver's fatigue state has strong practical significance.

The current mainstream driving fatigue detection methods are mainly divided into three categories: detection based on driver's physiological characteristics [3–5], detection based on motor vehicle behavior characteristics [6,7] and detection based on driver's facial feature [8–10].The detection method based on physiological characteristics is to understand the driver's fatigue state and judge the driver's fatigue degree through the collection, recording, detection and analysis of various physiological signals reflecting

driving fatigue. Although this method has high detection accuracy, it greatly reduces the driving experience. The second fatigue monitoring method is to measure the behavior of the vehicle. This method is mainly to monitor the changes of the data during the driving of the vehicle, such as speed, steering wheel rotation angle and lane departure detection, brake and accelerator pressure changes, etc., to indirectly evaluate the driver's fatigue through these data. This method is difficult to accurately judge the driver's mild fatigue. With the great success of deep neural networks in recent years for tasks such as speech recognition [11] and image classification [12], the rapid rise of neural network-based deep learning has provided new ideas for the development of fatigue driving systems. Zhao et al. [9] and Mohit et al. [10] have used methods that basically just extract features from the spatial dimension through a 2D convolutional operation kernel, but the limitation of this method is that none of them take into account the continuous inter-frame motion information. In a real driving environment, fatigue is usually a continuous state and extracting features in the spatial dimension without considering the temporal dimension will significantly compromise the detection results. The 3D convolution kernel [13] can extract temporal and spatial information in video data, so it can capture more motion information between video frames. After being proposed, 3D convolution is widely used in behavior recognition [14,15] and medical imaging [16,17], but is rarely used in fatigue detection. This paper proposes a fatigue detection model based on a 3D convolution kernel, which includes a face localization and tracking module, a 3D-CNN feature extraction module, a feature recalibration SE module and a fatigue state assessment module. The main contributions of this paper are as follows:

1.  In this study, 3D convolution and attention mechanism modules are used to extract features in both temporal and spatial dimensions and to recalibrate the fatigue features to improve the accuracy and robustness of the model.

2.  In this study, we conducted multiple sets of experiments on 3D-SE-Net to evaluate the detection effect of the model, including: comparative experiments with and without attention mechanism, comparative experiments for three-branch and double-branch networks, comparison experiments for different hyperparameters, comparison experiments for with and without frame aggregation strategy and comparison experiments for the proposed model with the state-of-the-art model on the same dataset.

3.  In this study, a dataset was collected in a real driving environment with a rich set of driving scenarios including day and night, with and without glasses and male and female, which can be used for subsequent in-depth studies on fatigue driving detection. In addition, an auxiliary software is proposed to record the changes of drivers' facial features in real time while protecting their facial privacy.

The remainder of this paper is organized as follows: Section 2 presents previous work on fatigue driving detection research. Section 3 expands the description of our proposed fatigue detection system. Section 4 introduces the relevant datasets and analysis of experimental results. Section 5 concludes the full text and looks forward to future work.

## 2. Related Works

The research on driver fatigue state detection can be traced back to the 1930s. Early researchers mainly tested drivers from the perspective of physiology and medicine, but the development was relatively slow. In the past two or three decades, with the development of sensors, image processing, machine vision, pattern recognition and other technologies, countries began to speed up the research work of driver fatigue state detection and made great progress. According to different fatigue monitoring methods, this section introduces some representative related methods: detection based on driver's physiological characteristics, detection based on motor vehicle behavior characteristics and detection based on driver's facial feature.

### 2.1. Detection Based on Driver's Physiological Characteristics

Khushaba et al. [3] proposed an efficient wavelet packet transform (FMIWPT) feature extraction method based on fuzzy mutual information (MI) to classify driver fatigue status and to extract fatigue-related information from EEG, EOG and ECG signals. Information and the quality of extracted features are evaluated based on simulation tests conducted on a data set collected from 31 drivers. The experimental results prove the key role of FMIWPT in feature extraction. The average accuracy of the experiment reached 97%. Chen et al. [4] also used a multi-feature fusion method to monitor drowsiness. First, the EEG signal was decomposed into wavelet sub-bands and nonlinear features were extracted from the fusion, then the eyelid movement information was fused with it and finally a support vector machine (SVM)was used for state classification. Fu et al. [5] studied the driver's fatigue state based on the EMG and ECG during driving and proposed a non-contact data acquisition system to collect the driver's physiological signals, using fast independent component analysis (FastICA) and digital filters to process the original Signal, according to the EMG crest factor and the maximum value of the cross curve of EMG and ECG to detect the driver's fatigue state. The results show that the proposed method can distinguish the normal state and the fatigue state well. It is worth mentioning that the proposed method is non-invasive, which helps to improve the stability of the detection results.

The method based on the driver's physiological characteristics is an accurate, effective and objective fatigue detection method, which is very convincing. Biological signs provide good signs of the early onset of fatigue and can be used to prevent accidents in time. However, this type of method is invasive and requires physical contact with the driver, which will greatly compromise the driving experience and cause inconvenience to the driver.

### 2.2. Detection Based on Vehicle Behavior Characteristics

The second fatigue monitoring method is to measure the behavior of the vehicle. This method is mainly to monitor the changes of the data during the driving of the vehicle, such as speed, steering wheel rotation angle and lane departure detection, brake and accelerator pressure changes, etc., to indirectly evaluate the driver's fatigue through these data.

Li et al. [6] calculated the approximate entropy (ApEn) based on the steering wheel angle (SWA) obtained in real time by the steering rod sensor of the car, used the given deviation for adaptive linear fitting and finally evaluated the driver's fatigue state according to the binary decision classifier. Only 78% accuracy was achieved in the car experiment. In recent years, based on the hybrid characteristics, the method of using the driver's physiological characteristics and vehicle behavior to monitor driving fatigue has become popular. To improve the robustness of fatigue detection, Sun et al. [7] proposed an adaptive dynamic recognition model that combines multiple information (blink frequency, closed eye time, vehicle turning angle deviation, etc.) and uses dynamic basic probability assignment (BPA) to realize the dynamic allocation of the weight of each information source. Experimental results show that information fusion can indeed improve the accuracy of fatigue detection, but there is still a certain gap with the real-world application standards.

This type of method is non-contact, but this method is greatly affected by driving conditions, driving experience, vehicle type, etc., and it takes a lot of time to collect and analyze the behavior of the driver and it is difficult to detect slight fatigue.

### 2.3. Detection Based on Driver's Facial Features

The detection method based on the driver's facial features refers to the analysis of the driver's face; it compares the driver's different facial expressions under fatigue conditions and normal conditions and summarizes some typical fatigue characteristics such as the driver's head posture, blink frequency, line of sight direction and yawn detection. According to these features, computer vision technology is used to detect and extract fatigue features from the driver's video image, calculate fatigue parameters and finally determine the driver's fatigue level. This detection method is low in cost, does not re-

quire contact and is convenient to detect; it is the popular research direction of driver fatigue detection systems.

Song et al. [8] used a Haar-based feature integrated detector combined with the intensity of multiple feature sets to characterize the structure and texture information of eye patches and proposed a new feature descriptor multi-scale principal gradient histogram (MultiHPOG), which can process the eye appearance in complex situations, improve the model's robustness to image noise and is of great significance for driving fatigue detection. With the development of deep learning, convolutional neural networks are widely used in image processing, target detection and other fields. Wang et al. [14] combined ResNet and DHLSTM to construct a driver fatigue detection system that copes with complex environmental disturbances. ResNet is used to extract driver facial features in space and CNN encodes the image sequences into feature vectors for input into a recurrent neural network LSTM for classification of driving status. Since the network structure of ResNet is relatively simple, it can achieve good real-time performance, but the performance of the network is more general for complex situations such as facial occlusion. Focusing on fatigue detection in terms of driver head pose estimation, Ansari et al. [15] proposed an improved bidirectional long- and short-term memory depth neural network for the analysis of head acceleration data in three time series. The driver's fatigue state was classified into three states: fatigue, normal and transition, and the overall accuracy of the training was 99.2%. This machine learning-based approach does not consider the detection of driver mouth features and eye features, which are the most obvious areas of driver fatigue signals. Ye et al. [16] proposed a residual channel-based attention network, which uses five-point localization to locate key parts of the driver's face and then uses RCAN to classify the state of the mouth and eyes. The channel attention module is added to RCAN, which improves at the same time. They also added a head pose estimation module using PnP [17] to evaluate whether the head pose is offset and finally achieved high detection accuracy on multiple datasets.

## 3. Materials and Methods

The overall structure of the driving fatigue detection system proposed in this paper includes the following parts: face tracking and privacy protection processing module, fully designed 3D-CNN module, Squeeze-and-Excitation module and a fatigue state analysis module.

### 3.1. Face Tracking and Privacy Protection Processing

When the driver's video is introduced into our fatigue detection system, the first thing we need to do is to detect and track the key points on the driver's face. We use the Dlib [18] library to detect and track the key areas of the face image. Dlib is a platform-independent programming library whose source code is based on a C++ implementation, but which provides a Python interface. It is more versatile and superior in image processing and facial feature extraction, classification and comparison. The Dlib library provides a facial landmark detector containing 68 coordinate points of a human face. The Dlib library detects and tracks the driver's face in real time and at the same time accurately obtains the driver's key point coordinates. According to the coordinate output of the feature points, the micro movements of the human face can be accurately tracked.

It is worth mentioning that a Dlib library-based application was designed to monitor the driver's face and head features in real time while protecting the driver's facial privacy. The driver's blinking, nodding and yawning behaviors are monitored and recorded in real time. The interface of the software is shown in Figure 1. The video recording device is located on the upper part of the vehicle's front windshield and can record the driver's facial changes in real time without affecting the driver's normal driving. For the processing of driver's facial privacy protection, the method of template fusion is used to fuse face features. First, we obtained a face template database through web crawling and dataset integration. After that, we classified the attributes of faces (e.g., age, expression, skin quality,

skin tone, angle, etc.,) and after the driver's face enters the collection range, the model will traverse the database to match templates with similar attributes to the driver's image; the matching algorithm is shown below.

$$y = \operatorname*{argmin}_{\text{Lab}} \left( \begin{array}{c} (O_1 - M_1)^2 \times \theta_1 + (O_2 - M_2)^2 \times \theta_2 + \\ (O_3 - M_3)^2 \times \theta_3 + \ldots (O_n - M_n)^2 \times \theta_n \end{array} \right) \tag{1}$$

where Lab denotes the template library, $O_{1\sim n}$ denotes the original image face attributes, $M_{1\sim n}$ denotes the matching template image attributes and $\theta_{1\sim n}$ denotes the coefficients of the corresponding attributes. After matching the most suitable face template, we have to segment the face. We segment the face region based on the key points of the face identified by Dlib using the Delaunay triangulation algorithm [19]. The Delaunay triangulation algorithm can segment the points on the plane into triangles with good properties. The segmentation of the template image and the original image is consistent, which facilitates the fusion of facial key points afterwards. The generation of the fused image is based on the following equation:

$$I_p = (1 - \xi)I_o + \xi I_m \tag{2}$$

where $I_p$, $I_o$ and $I_m$ represent the fused face image, the original image and the matched image in the template library, respectively. $\xi$ represents the deformation intensity, which is an artificially set parameter greater than 0 and less than 1. Since the key regions for fatigue detection are eyes and mouth, the weights in these two regions are set to 0 to ensure the accuracy of fatigue detection. The fused images are presented in the software interface after 3D simulation as shown in Figure 2, which solves the problem of driver facial information leakage from the bottom and better ensures the security of driver facial privacy.

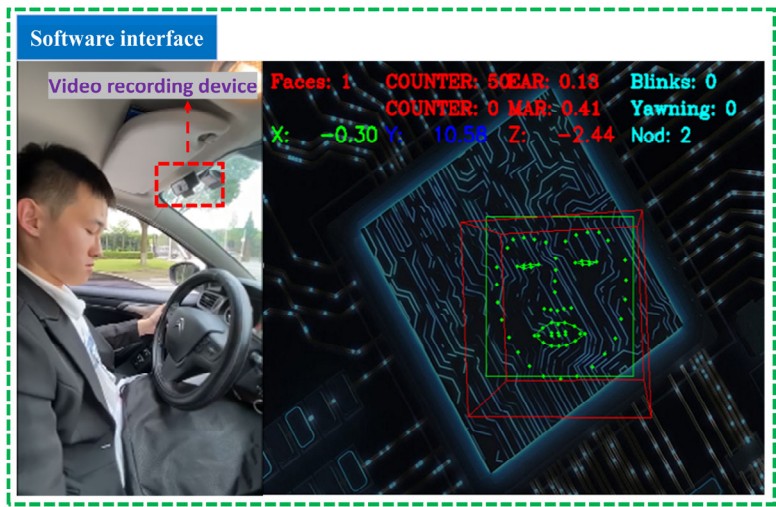

**Figure 1.** Fatigue driving detection software interface.

In the actual driving environment, the phenomenon of missing detection of the driver's facial features (camera occlusion, driver nodding, etc.) may occur. In order to ensure the accuracy of detection effect and maximize driver safety, frame aggregation is used to overcome the phenomenon of missing detection. In a certain consecutive frame, if there is no face detected in a certain frame, an adjacent frame is used to replace the image of this frame so as to ensure that the driver's face image is always clearly detected in every frame of the video stream. The specific operation principle of this strategy is shown in Figure 3.

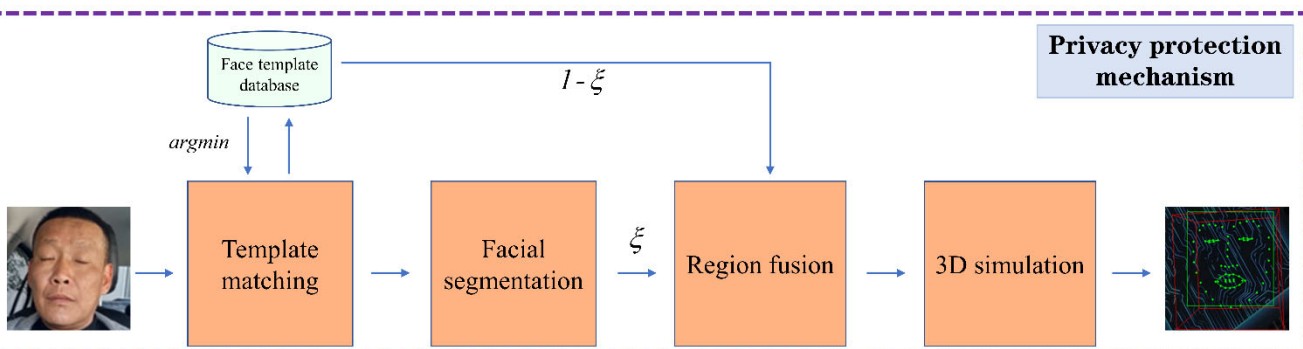

**Figure 2.** Schematic diagram of privacy protection mechanism.

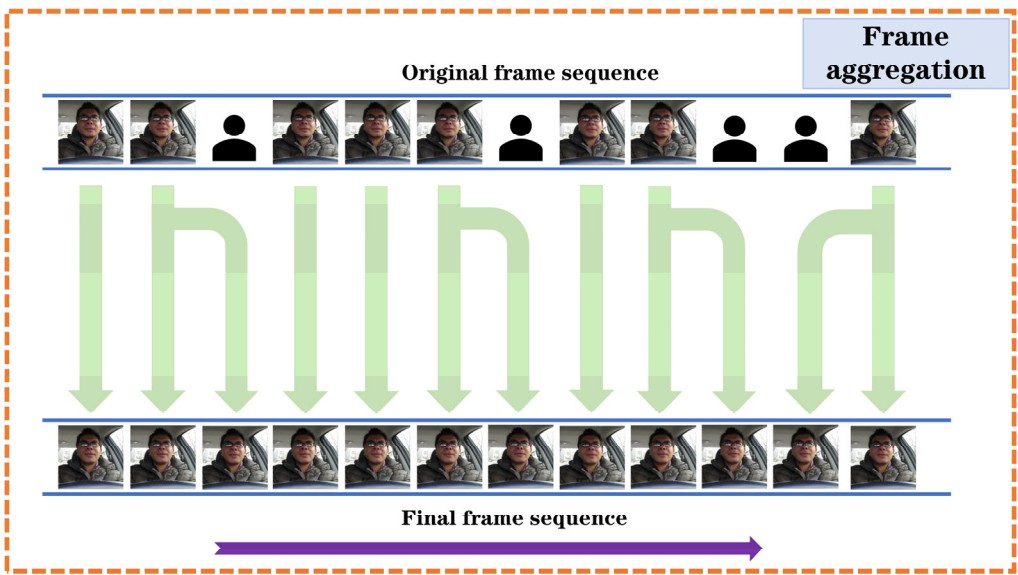

**Figure 3.** Schematic diagram of frame aggregation strategy.

### 3.2. 3D-CNN Module

In the following, we describe our 3D-SE-Net fatigue detection architecture in depth. As shown in Figure 4a, the proposed framework consists of three branches, each with the same convolution and pooling operations, which are trained on images of the driver's left eye, right eye and mouth. After acquiring the grey-scale, gradient and optical flow images of the driver's video frame, the feature maps are passed into the corresponding 3D-CNN branches, each containing two 3D convolution operations and two max-pooling operations. The feature maps of each branch are merged together to generate a feature map cube and finally to enter the SE module [20] to compress and excite the features on the feature channels and then re-assign the weights to each channel to obtain the final fatigue recognition result. In order to prevent the gradient from disappearing and to keep the convergence rate of the model in a steady state, each 3D convolution operation is followed by a linear rectification function (ReLU) [21] as the activation function, which facilitates the fitting of the training data while preventing overfitting to a certain extent.

The parameters and structure of each 3D-CNN branch are shown in Figure 4b. First, in the input layer, we select 10 consecutive frames of the driver's face of size $48 * 48$ from the video stream as the input to the 3D-CNN network and then obtain an image containing five channels of information from the original image of the driver in the hardwired layer, following the method used by Ji et al. [13] to encode the original image with optical flow and gradient features. They are greyscale, optical flow-x, optical flow-y, gradient-x and gradient-y. Of these, the greyscale and gradient images we can obtain directly by frame-

by-frame computation, whereas the optical flow image is special in that it contains motion information between consecutive frames, so it must be obtained by two consecutive frames. Thus, in the hardwired layer we get 48 frames $(10 * 3 + (10-1) * 2 = 48)$ of the driver's face image containing 5 channels of information. These sample images from different channels actually encode our prior knowledge of the features, which helps to improve the convergence and generalization of the network.

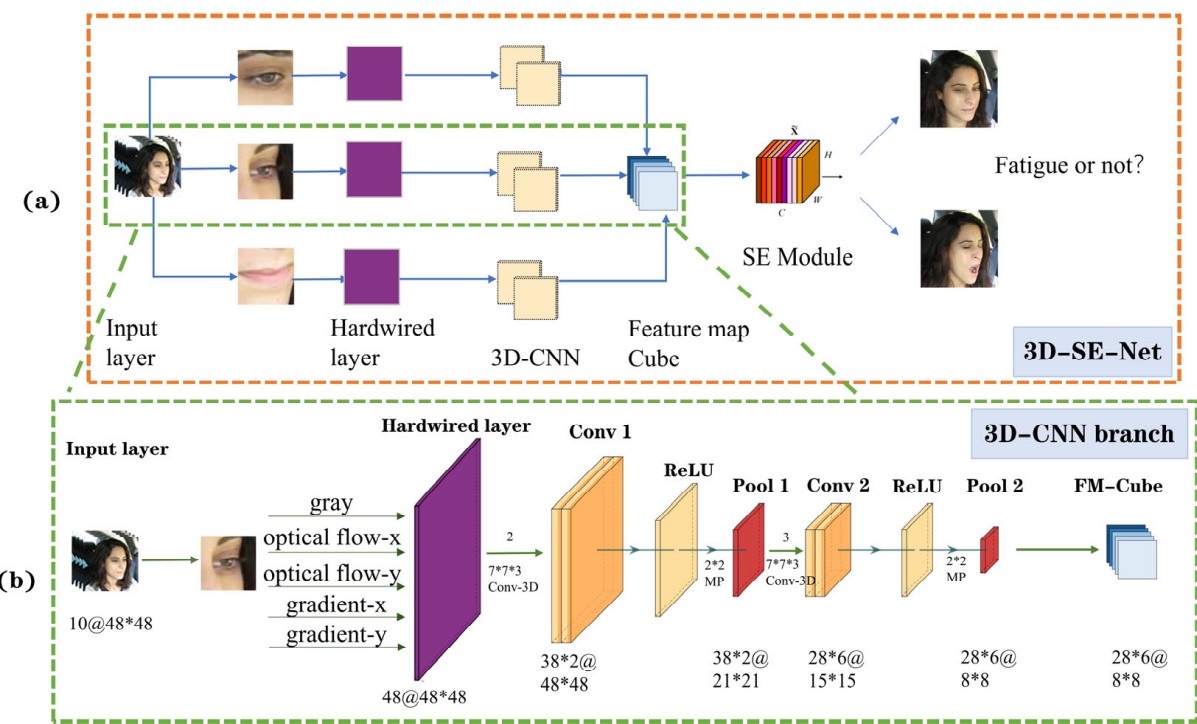

**Figure 4.** (**a**) The proposed 3D-SE-Net framework; (**b**) detailed diagram of the structure and parameters of the network layers.

The first 3D convolution layer, Conv1, contains two sets of 3D convolution kernels of size $7 * 7 * 3$, where $7 * 7$ denotes the spatial dimension and 3 denotes the temporal dimension, so that we obtain two sets of feature maps of size $42 * 42$ and 38 channels. After activation by the ReLU function, we apply the max pooling layer Pool1 with the kernel size of $2 * 2$ to downsample the feature maps, which maintains the number of feature maps but decreases the spatial resolution. To enhance the number of feature maps, we employ three sets of convolution kernels of size $7 * 7 * 3$ for the second 3D convolution operation, yielding six sets of feature maps of channel number 28 and size $15 * 15$. After activation by the ReLU function into the second maximum pooling layer Pool2, this layer has exactly the same parameters as the Pool1 layer, so we get a feature map with no change in the number of channels and a size of $8 * 8$.

After two convolution and pooling operations, we stack the feature maps of the three tributaries according to the channel arrangement and transform the 10 input images of size $48 * 48$ into a feature map cube of $8 * 8 * 504$. Subsequently, the feature map cube is sent to the SE module for weight optimization and reassignment; the specific operations are described in Section 3.3 in the expansion. Finally, we use stochastic gradient descent(SGD) [22] to train the whole network model.

### 3.3. Squeeze-and-Excitation Module

After feature extraction by 3D-CNN, the feature map contains rich spatial and channel information and then we use the Squeeze-and-Excitation (SE) module [20] to perform 'feature rescaling' on the feature map. The SE module is not a complete network but a module for feature learning and redistribution. The common convolutional neural network

is usually an information aggregator that aggregates spatial information and channel information on the local receptive field, while the SE module adopts a strategy of 'feature rescaling' to explicitly model the interdependence between feature channels. To be specific, it is to automatically acquire the importance of each channel through learning, and then promote useful features according to this importance and suppress features that are not useful for the current task. In our proposed 3D-SE-Net, the driver's facial features are reconstructed using the weight matrix learned by the SE module, which has two main operations: Squeeze and Excitation, as described below.

### 3.3.1. Transformation and Squeeze Operation

Before performing the Squeeze operation, in order to make the size and number of channels of the feature map more compatible with the output, we first perform a feature transformation on the feature map cube, for a given input: $\mathbf{X} \in \mathbb{R}^{H' \times W' \times C'}$, and let it do a mapping $\mathbf{F}_{tr}$ to get an output: $\mathbf{U} \in \mathbb{R}^{H \times W \times C}$, where

$$\mathbf{U} = \mathbf{F}_{tr}(\mathbf{X}) \tag{3}$$

In a traditional CNN, this step is actually an ordinary convolution operation.

The next key step of our SE module is to use global average pooling (GAP) to compress the corresponding spatial information ($H \times W$) on each channel into a value in the corresponding channel, where a pixel represents a channel and finally becomes a vector of dimension ($1 \times 1 \times C$). The specific operation is as follows: after the first step of the transformation of the matrix X by $\mathbf{U}$ to do a global average pooling (GAP), to get $\mathbf{Z}$, where

$$\mathbf{Z} = \mathbf{F}_{sq}(\mathbf{U}) = \frac{1}{H \times W} \sum \sum (i, j) \tag{4}$$

Through the Squeeze operation, a global low-dimensional embedding is achieved for $\mathbf{U}$. As shown in Figure 5, the spatial information is squeezed so that the values obtained from each channel after compression have a global receptive field.

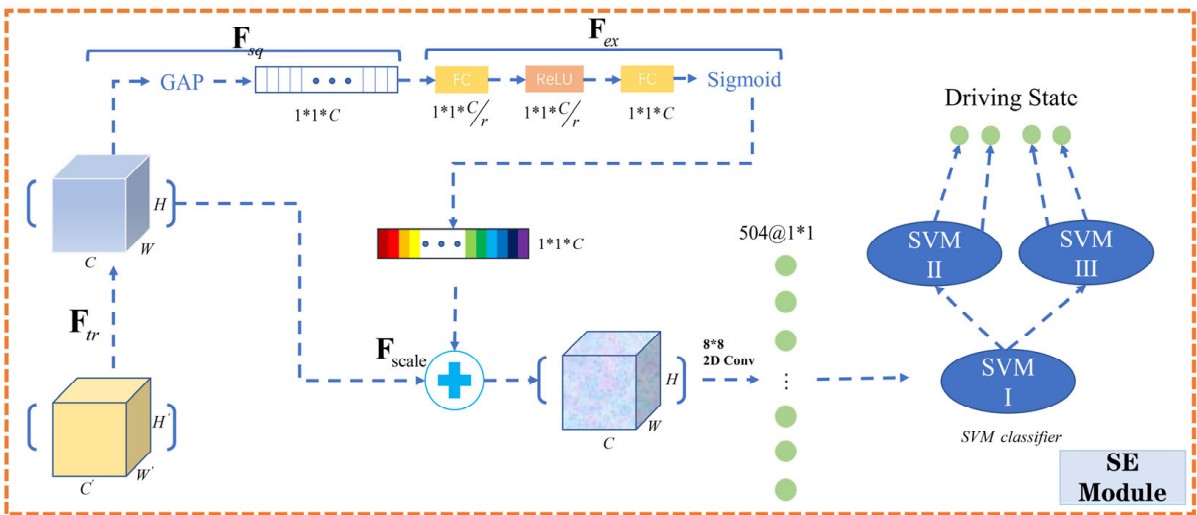

**Figure 5.** Attention mechanism SE module.

### 3.3.2. Excitation and Scale Operation

In order to take full advantage of the aggregated information obtained by spatial compression, the Excitation operation is engaged in the next stage. Since the previous Squeeze operations are only performed in the feature maps of the respective channels, we next need to establish the dependencies between the channels, so we use the fully

connected layer of the bottleneck structure to fuse the feature map information of each channel as follows:

$$\mathbf{s} = \mathbf{F}_{ex}(\mathbf{Z}, \mathbf{W}) = \sigma(g(\mathbf{Z}, \mathbf{W})) = \sigma(\mathbf{W}_2 \delta(\mathbf{W}_1 \mathbf{Z})) \tag{5}$$

where $\mathbf{W}_1 \in \mathbb{R}^{\frac{C}{r} \times C}$ and $\mathbf{W}_2 \in \mathbb{R}^{C \times \frac{C}{r}}$ are the corresponding parameters of the two fully connected layers. Using two fully connected layers increases the nonlinearity of the network while improving the fitting ability. $\delta$ is the activation function ReLU [21] and $\sigma$ is the activation function Sigmoid. The values of each channel output after the activation function Sigmoid belong to 0~1. These values are also the weight coefficients of each channel, so that the utilization of useful channel information and the suppression of useless channel information are achieved. It is worth mentioning that there is a hyperparameter $r$ in this operation and we will develop a description of the effect of different $r$ values on the recognition accuracy in Section 4.5.

In the next Scale operation, we apply the obtained weights to each channel above $\mathbf{U}$. Specifically, for all the values on $H \times W$ at each position of $\mathbf{U}$, we multiply the weights of the corresponding channels, completing the recalibration of the feature map.

$$\widetilde{\mathbf{X}} = \mathbf{F}_{\text{scale}}(\mathbf{U}, \mathbf{s}) = S * U \tag{6}$$

After the 3D-CNN and SE modules, we used an $8 * 8$ 2D convolution operation to obtain 504 feature vectors of size $1 * 1$ to capture the motion information contained in the input frames. We finally combined multiple binary classification SVMs to achieve the classification of fatigue driving behavior.

### 3.3.3. Driver Behavior Analysis

When the eyes and mouth are open, their aspect ratio is usually constant, while when they are closed the value of the aspect ratio is close to 0. This ratio does not change with individual human differences. Therefore, in the proposed model, we use EAR [23] and MAR [24] to assess the fatigue state of the driver.

In the research of Tereza et al. [23], the concept of EAR was first proposed. The obtained feature map contains six key points of the human eye as shown in Figure 6. When the person blinks, the distance of these six points will change; the distance relationship of these six points can be used to judge whether there is blinking behavior and the length of blinking time can be judged by the number of frames. Similarly, we use a similar algorithm to evaluate mouth opening and closing (MAR). To this end, we designed a full binary tree SVM classifier to classify the states of eyes and mouths and the final outputs include: (a) closed eyes; (b) yawning; (c) sleepy-nodding. To speed up the efficiency, we use the strategy of transfer learning [25] to pre-train the SVM on the SMIC database [26] (the Spontaneous Micro-expression database). The SMIC database contains a large number of images of human facial expressions and the pre-trained models can be directly used for our classification task after fine-tuning.

We use the aspect ratio of the eyes to evaluate the opening and closing of the eyes. The formula for the aspect ratio of the eyes is as follows:

$$\text{EAR} = \frac{\|P_2 - P_6\| + \|P_3 - P_5\|}{2\|P_1 - P_4\|} \tag{7}$$

Likewise, the formula for the aspect ratio of the mouth looks like this:

$$\text{MAR} = \frac{\|P_{51} - P_{59}\| + \|P_{53} - P_{57}\|}{2\|P_{49} - P_{55}\|} \tag{8}$$

where $P_i$ represents the position of the key points of the eyes and mouth. In [23], the aspect ratio threshold of the eyes is set to 0.2 and in [24] the aspect ratio threshold of the mouth is set to 0.8. In this model, two thresholds are obtained experimentally.

The average blink time for a normal person is 0.2–0.4 s, while the blink time in a fatigued state usually lasts for 2 s. The value of MAR is always changing during normal speech and if the value of MAR is stable and consistently exceeds the threshold, we assume that the driver has made a yawning movement. Therefore, we can determine whether the driver is in a fatigue state based on the number of frames in the video. If the number of frames with blink time longer than 2 s exceeds 60%, or the number of frames with yawn exceeds 40%, we assume that the driver is in a fatigue state.

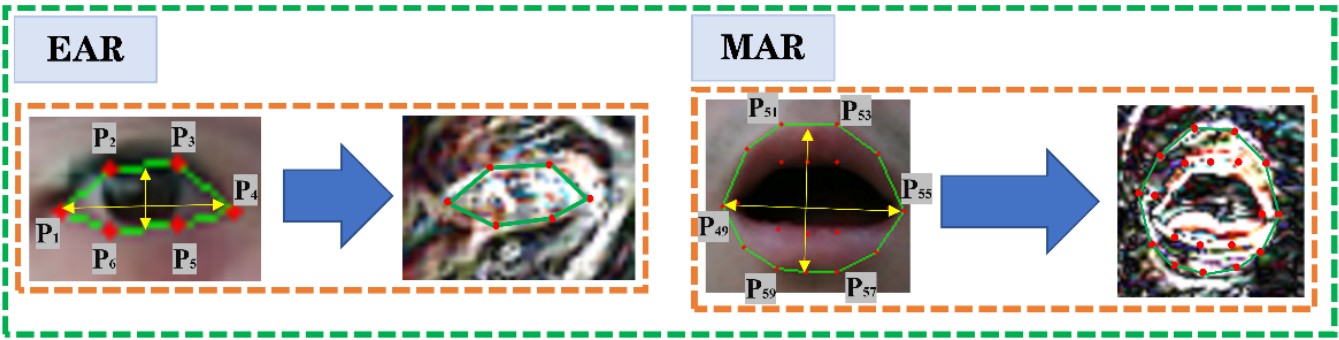

**Figure 6.** Schematic diagram of EAR and MAR.

## 4. Experiments and Result Analysis

This section may be divided by subheadings. It should provide a concise and precise description of the experimental results, their interpretation, as well as the experimental conclusions that can be drawn.

### 4.1. Implementation Details

The experiments were conducted on a NIVIDA GeForce RTX2070 graphics board, using the Pytorch deep learning framework to build the network and CUDA and CUDNN provided by NVIDIA to train and test the dataset respectively. The experimental input image size is 48 ∗ 48 and the different scale features were obtained through 3D convolution, then entered in the SE module to learn the importance of different channel features and finally obtaining the fatigue state classification results by full binary tree SVM classifier. We trained the model for 100 epochs and the model was trained using stochastic gradient descent (SGD) with an initial learning rate of 0.001 and a momentum of 0.9 during training.

### 4.2. Dataset

We used four datasets to train our proposed 3D-SE-Net and validated the performance of our proposed model. The datasets we used for network training are described in the following. Some examples in the dataset are shown in Figure 7.

ZJU Dataset [27]: The first dataset contains videos of twenty individuals with four clips each, containing a total of 80 videos of subjects in four categories: frontal without glasses and frontal with light-colored glasses, frontal with dark-colored glasses and supine without glasses, with the main subjects in the dataset being of Asian origin. Zhao et al. [9] performed geometric transformations, Gaussian blurring and other graphical transformations on the ZJU dataset to reduce the resolution of the images. Their strategy was adopted in our approach.

NTHU dataset [28]: The second dataset contains driving recordings of 36 subjects of different races in five different scenes (BareFace, Glasses, Night_BareFace, Night_Glasses, Sunglasses), including normal driving, yawning, slow blinking, falling asleep, etc. The scenes contain both day and night lighting states, the video resolution is 640 × 480 in AVI format and we converted the video to a picture of size 48 ∗ 48, containing the state of glasses and mouth.

YawDD dataset [29]: The third dataset is video data recorded by in-car cameras, recording various facial features of drivers in a real car (male and female, with or without

glasses/sunglasses, different ethnicities), talking, singing, remaining silent and yawning. We transformed all 351 videos into picture frames of size 48 ∗ 48, including 5435 pictures of the mouth state and 6120 pictures of the eyes state, which were divided to scale for model training and validation.

FDF dataset: The fourth dataset was designed and collected fully in-house and we have named it the FDF dataset. This dataset contains a total of 16 people, both with and without glasses, and the test environment includes both daytime and nighttime scenarios. To ensure the objectivity of the data, our dataset was collected in a real driving environment. The subjects' driving states included 'Normal', 'Talking', 'Sleepy-nodding' and 'Yawning', and we eventually acquired 5360 images of the driver's facial state from the video frames.

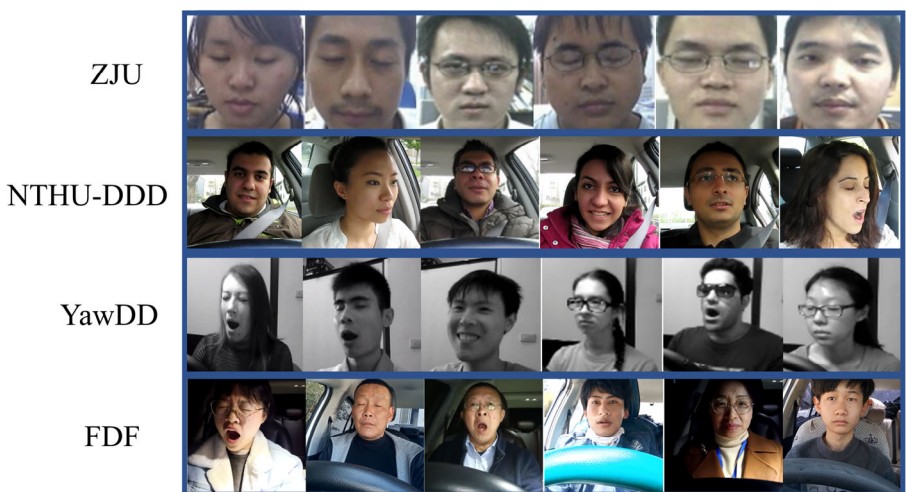

**Figure 7.** Some samples of the fatigue driving dataset.

### 4.3. Performance of 3D-SE-Net versus Other Network

To demonstrate the capability of our proposed model in driving fatigue detection, we conducted a large number of comparative experiments on four datasets. The recognition results of the current mainstream networks including ResNet-50 [30], MobileNetV3 [31] and EfficientNet [32] were compared. Moreover, we simplified the proposed network with the main network containing three input branches including the right eye, the left eye and the mouth (Three-branch 3D-SE-Net). We modified the network to input only a single human eye and mouth (Double-branch 3D-SE-Net) as a set of comparison experiments. The specific experimental results are shown in Table 1.

**Table 1.** Classification accuracy of the proposed method and different classical CNN structures on four datasets (where T stands for Three-branch 3D-SE-Net and D stands for Double-branch 3D-SE-Net).

| Dataset | Training Object | Test Data | Method | Accuracy (%) |
|---------|-----------------|-----------|--------|--------------|
| *ZJU* | Eyes | 975 | ResNet-50 | 95.64 |
| | | | MobileNetV3 | 95.73 |
| | | | EfficientNet | 96.22 |
| | | | 3D-SE-Net(T) | **98.35** |
| | | | 3D-SE-Net(D) | **97.72** |
| | Mouth | 634 | ResNet-50 | 95.65 |
| | | | MobileNetV3 | 96.17 |
| | | | EfficientNet | 96.41 |
| | | | 3D-SE-Net(T) | **97.17** |
| | | | 3D-SE-Net(D) | **96.84** |

**Table 1.** *Cont.*

| Dataset | Training Object | Test Data | Method | Accuracy (%) |
|---|---|---|---|---|
| NTHU-DDD | Eyes | 1135 | ResNet-50 | 97.31 |
| | | | MobileNetV3 | 97.47 |
| | | | EfficientNet | 97.72 |
| | | | 3D-SE-Net(T) | **99.02** |
| | | | 3D-SE-Net(D) | **98.53** |
| | Mouth | 990 | ResNet-50 | 97.09 |
| | | | MobileNetV3 | 97.43 |
| | | | EfficientNet | 98. 27 |
| | | | 3D-SE-Net(T) | **98.84** |
| | | | 3D-SE-Net(D) | **98.31** |
| YawDD | Eyes | 2014 | ResNet-50 | 97.13 |
| | | | MobileNetV3 | 97.26 |
| | | | EfficientNet | 97.48 |
| | | | 3D-SE-Net(T) | **98.67** |
| | | | 3D-SE-Net(D) | **98.04** |
| | Mouth | 1065 | ResNet-50 | 97.32 |
| | | | MobileNetV3 | 97.47 |
| | | | EfficientNet | 97.72 |
| | | | 3D-SE-Net(T) | **98.55** |
| | | | 3D-SE-Net(D) | **97.81** |
| FDF | Eyes | 5461 | ResNet-50 | 98.45 |
| | | | MobileNetV3 | 98.61 |
| | | | EfficientNet | 98.57 |
| | | | 3D-SE-Net(T) | **99.12** |
| | | | 3D-SE-Net(D) | **98.74** |
| | Mouth | 2136 | ResNet-50 | 98.32 |
| | | | MobileNetV3 | 98.57 |
| | | | EfficientNet | 98.62 |
| | | | 3D-SE-Net(T) | **99.38** |
| | | | 3D-SE-Net(D) | **98.55** |

As can be seen from Table 1, the proposed 3D-SE-Net performs significantly better than other networks on the ZJU dataset, due to the proposed model extracting features in both temporal and spatial dimensions through 3D convolution, capturing motion information from multiple consecutive frames and further improving performance compared with ordinary 2D convolution. Compared with other datasets, the ZJU dataset contains images with lower resolution and the final experimental results demonstrate that the proposed model still has better robustness in processing low-resolution images, with 3D convolution outperforming 2D convolution for the same dataset. On both the YawDD dataset and the NTHU dataset, the proposed model achieves high recognition accuracy compared with other models because the images contained in these two datasets are mostly frontal images of drivers, which are easy to recognize. On the FDF dataset, the accuracy of each model is improved to some extent as the dataset contains sufficient images and the models are well fitted under this dataset. We can also find that the recognition of Three-branch 3D-SE-Net is better than that of Double-branch 3D-SE-Net under the same dataset, which indicates that the proposed model strategy is successful.

### 4.4. Evaluation of the Impact of the Attention Mechanism on the Model

To verify the improvement of the attention mechanism module on the performance of the proposed network, we modified the base 3D-SE-Net by removing the SE module and using a linear classifier to output detection results as mentioned in [13]. After obtaining a feature map of size $8 * 8$ and $28 * 6$ channels in the Pool2 layer, the feature map is converted into feature vectors using $8 * 8$ 2D convolution. These feature vectors are output directly

through SVM classifier without going through the attention mechanism module. This model is named '3D-CNN' in Table 2.

In this experiment, based on the FDF dataset, we retrained the two models on this dataset for 80 epochs each. To ensure objectivity, we used the exact same images for training. It is worth mentioning that we introduced confusion matrices to visualize the detection results and also used precision, recall and F1-score as metrics to judge the accuracy of the models. Precision refers to the proportion of positive samples judged by the classifier as positive cases, recall is the proportion of positive cases predicted to be positive to the total number of positive cases and F1-score is a measure of the classification problem that is the summed average of precision and recall, with a maximum of one and a minimum of zero. A total of 1302 images were used as the test set, including 487 'Normal' images, 283 'Talking' images, 152 'Sleepy-nodding' images and 380 'Yawning' images. The performance of these two models on the test set is shown in Figure 8.

By analyzing the two confusion matrices in Figure 8, we find that the main diagonal region of the confusion matrix is darker, while the region outside the main diagonal is lighter. This indicates that the proposed 3D convolution operation is effective for the detection of key regions of the driver's face and that the optical flow and gradient encode the original features, bringing good results. However, in Figure 8 (Left), it is clear that the 3D-CNN has a large deficit in the classification of 'Talking' and 'Yawning'. Both behaviors are very similar and both include an act of opening the mouth. Due to the different positions of the people in the dataset, the different mouth sizes and the camera angles, it is inevitable that the two behaviors are defined with a high degree of error. The 3D-CNN achieves good results for the two behaviors 'Normal' and 'Sleepy-nodding'. With the addition of the attention mechanism SE module (3D-SE-Net), the recognition rate of the two driving behaviors 'Normal' and 'Sleepy-nodding' reached almost 100%, as shown in Figure 8 (Right). The recognition of the 'Talking' and 'Yawning' behaviors is also a significant improvement over the original 3D-CNN. We calculated the precision, recall and F1-Score of the two models on the test set, as shown in Table 2.

The table more clearly reflects the improvement in model performance by the attention mechanism SE module compared with the confusion matrix, with 3D-SE-Net achieving an average precision and recall of 95% for the recognition of these four driving behaviors. In the 3D-CNN model, the F1-Score value for the behavior 'Talking' was only 74%, while in the 3D-SE-Net model it increased to 92%, and the F1-Score values for 'Sleepy-nodding' and The F1-Score for 'Sleepy-nodding' and 'Yawning' also increased by more than 10 percentage points, suggesting that the attention mechanism can be effective in the recognition of various driving behaviors.

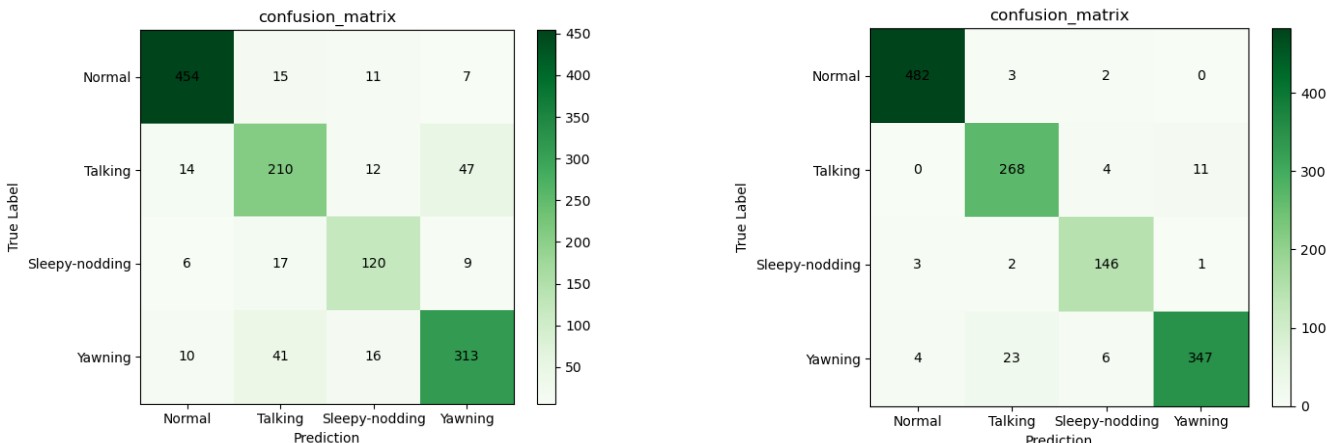

**Figure 8.** Confusion matrix of 3D-CNN on FDF test set (**Left**); confusion matrix of 3D-SE-Net on FDF test set (**Right**).

**Table 2.** Performance evaluation results on the FDF test set.

| Model | 3D-CNN (Without Attention) | | | 3D-SE-Net (With Attention) | | |
|---|---|---|---|---|---|---|
| | P | R | F1 | P | R | F1 |
| Normal | 0.94 | 0.93 | 0.95 | 0.99 | 0.99 | 0.99 |
| Talking | 0.74 | 0.74 | 0.74 | 0.91 | 0.94 | 0.92 |
| Sleepy-nodding | 0.75 | 0.79 | 0.77 | 0.92 | 0.96 | 0.94 |
| Yawning | 0.83 | 0.82 | 0.82 | 0.97 | 0.91 | 0.94 |
| Average | 0.82 | 0.82 | **0.82** | 0.95 | 0.95 | **0.95** |

To show more intuitively the effect of the attention mechanism on the optimization of the model, we visualized the feature maps using Grad-CAM [33]. Instead of segmenting the face image, we retooled the network to feed the whole face image into both models, which was done to show our visualized results more clearly. As can be seen in Figure 9, the 3D-SE-Net with the attention mechanism module added maintains good detection in both bright and dark lighting conditions, while the 3D-CNN without the attention mechanism performs relatively poorly in dark conditions. In addition, the heat map shows that the facial detection under the condition of wearing sunglasses is not satisfactory with or without the attention mechanism module. We analyze that this is due to the lack of training data for the model and the small number of sunglasses in the FDF dataset makes the 3D convolutional neural network overfitting. The model cannot accurately identify where the eyes are located, which is detrimental to the use of EAR. For this phenomenon using infrared camera to collect data may be a better solution. In the control group with and without glasses, the recognition effect of both models decreases to some extent, but 3D-SE-Net still outperforms 3D-CNN. Thus, we conclude that both lighting conditions and facial occlusion affect the recognition effect of the models to different degrees.

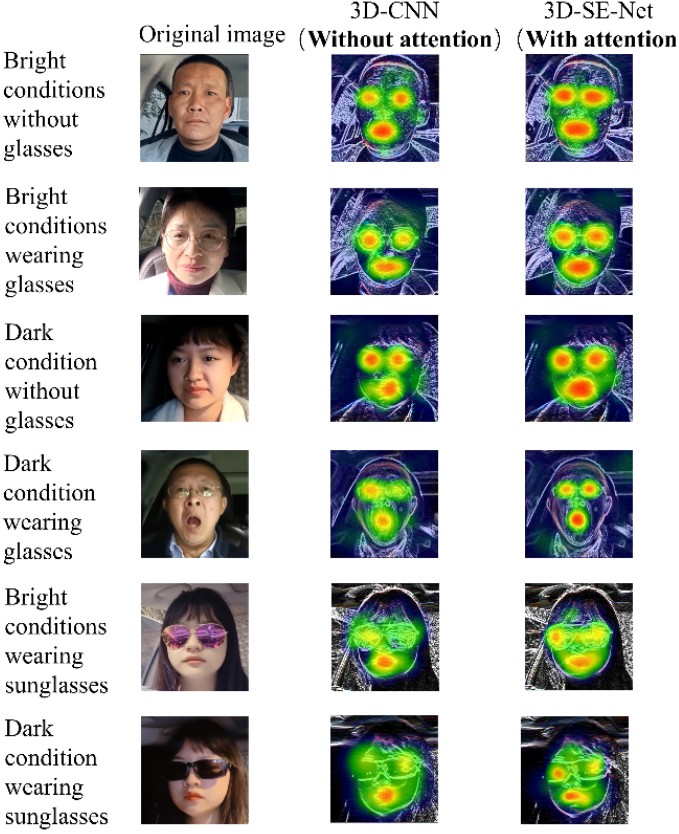

**Figure 9.** Feature map processing results for different drivers and driving environments using Grad-CAM.

*4.5. Evaluation of the Impact of Different Hyperparameter r Values on the Model Accuracy*

In the 'Excitation' operation of the SE module, two fully connected layers are included and the size of the feature map is transformed from $1 * 1 * C$ to $1 * 1 * C/r$ through a bottleneck structure, and finally back to $1 * 1 * C$. The value of the hyperparameter $r$ will affect the parameter quantity and accuracy of the model. We conducted a series of ablation experiments for the hyperparameter $r$ based on the YawDD dataset.

As can be seen from Figure 10, we have set a total of four sets of $r$ values for parallel experiments. On the 3D-SE-Net model, with the increase of $r$ value, the detection accuracy of each fatigue behavior is gradually improved, but in the 3D-SE-Net model, after the $r$ value exceeds 16, there is a large drop in accuracy, which indicates that the network may be overfitting. The highest accuracy is achieved when $r$ is 16, which is also used in our model.

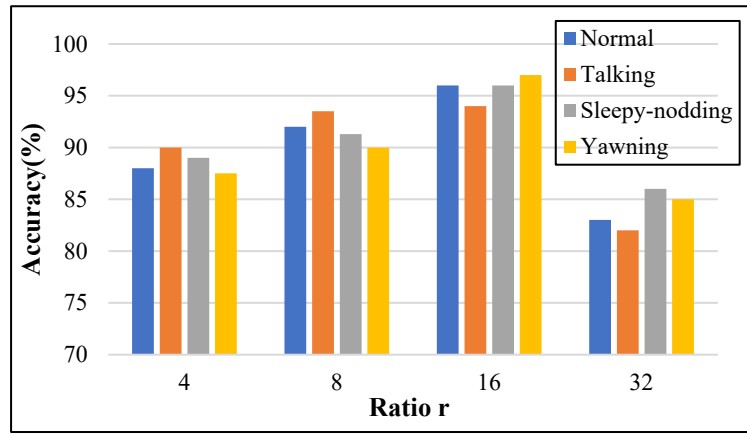

**Figure 10.** The detection accuracy of each behavior under different hyperparameter $r$ values.

*4.6. Evaluating the Impact of Frame Aggregation Strategy on Model Accuracy*

In this section, 3D-SE-Net models with and without frame aggregation strategy are compared. To better reflect the role of the frame aggregation strategy, we use images of driving scenes in dark environments as our test set and the images in the test set are all derived from the FDF dataset. We use the same dataset to train both models and guarantee the same training epoch.

As can be seen from Figure 11, under dark conditions, the recognition accuracy of the group with frame aggregation strategy is significantly higher than that of the group without. The purpose of the frame aggregation strategy is to prevent the loss of video frames, which is difficult to avoid in the dark driving environment with poor lighting conditions. Judging from the recognition accuracy of each driving behavior, the recognition accuracy of each driving behavior is increased by 4~7 percentage points, which shows that the frame aggregation strategy can well improve the problem of poor lighting conditions.

*4.7. Comparison of the Proposed Method and State-of-the-Art Methods*

In this section, we compare the proposed 3D-SE-Net with state-of-the-art fatigue detection methods. In order to ensure the objectivity of the experiments, the methods we compared, Zhang et al. [34], Deng et al. [35], Ji et al. [36], Ye et al. [16], are all based on the YawDD dataset. We train both Double-branch 3D-SE-Net (D) and Three-branch 3D-SE-Net (T). A 300-frame test set is constructed based on the YawDD dataset, the hyperparameter $r$ is 16 and the driver's fatigue state is obtained through the SE module. The experimental results are shown in Table 3 and the method proposed in this paper outperforms the current state-of-the-art methods.

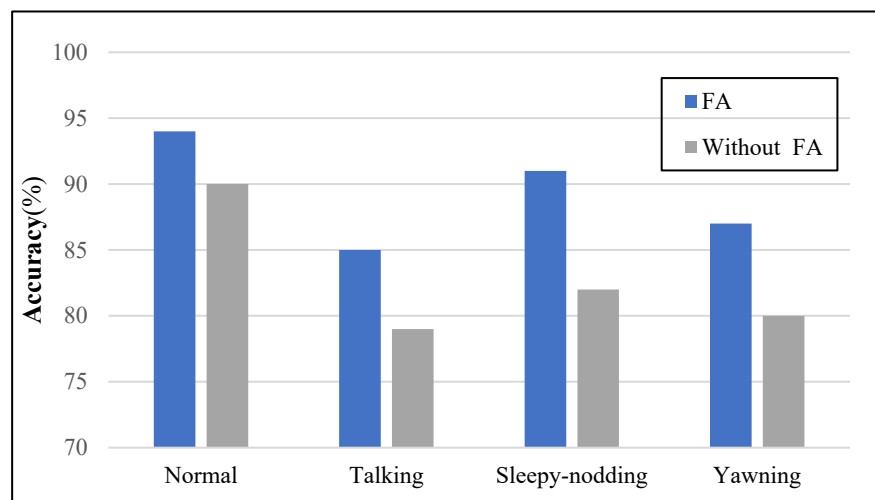

**Figure 11.** Comparison of the accuracy of each behavior detection with or without frame aggregation strategy on the FDF dataset.

**Table 3.** Comparison results of the proposed method and state-of-the-art methods.

| Research | Dataset | Method | Accuracy (%) |
|---|---|---|---|
| Zhang et al. [34] | YawDD | Kalman + TLD + CNN | 92 |
| Deng et al. [35] | YawDD | MC-KCF + blinking + yawning | 96.3 |
| Ji et al. [36] | YawDD | MSR-Net | 98.45 |
| Ye et al. [16] | YawDD | RCAN(CBAM) | 98.43 |
| Wang et al. [14] | NTHU | Resnet + DHLSTM | 99.3% |
| Ansari et al. [15] | Homemade | reLU-BiLSTM | 99.2% |
| Ours | YawDD | 3D-SE-Net(D) | 97.34 |
| | YawDD | 3D-SE-Net(T) | **99.03** |

Wang et al. [14] combined the residual network ResNet and the recurrent neural network DHLSTM to construct a driver fatigue detection system that copes with complex environmental disturbances. The introduction of recurrent neural network can solve the problem of gradient disappearance and gradient explosion during the training of long sequences, with efficiently fatigue detection accuracy, while the simple structure of the residual network makes the model work well in real time. The temporal problem is also taken into account in our model, and from the experimental results, the recognition accuracy of both models is approximately the same on the same dataset. Ansari et al. [15] proposed an improved bidirectional long and short-term memory deep neural network for the analysis of head acceleration data in three time series. They designed a new classifier called asda, where all fatigue features were determined by the head pose. Although the authors achieved a high accuracy in their experiments, we believe that better detection would be achieved if eye and mouth features were added to the model. It is worth mentioning that the RCAN proposed by Ye et al. [16], a model that also includes an attention mechanism module (CBAM), uses the residual network to extract features of the human eye and mouth and the attention mechanism module to extract semantic information of the image, achieving good classification results. The attention mechanism allows the network to adaptively learn 'Which part is more important?' In our model, the 3D-SE-Net with the attention mechanism module also performs well, achieving an accuracy of 99.03% in the recognition of fatigue states.

## 5. Discussion

In this paper, we propose a driving fatigue detection model based on a 3D convolutional neural network combined with a channel attention mechanism module. The experiments are based on four datasets, and multiple sets of control experiments are

conducted for different lighting and occlusion conditions, with or without the attention mechanism module, and whether to use the frame aggregation strategy. It is compared with other current fatigue detection methods and achieves satisfactory detection accuracy. Unlike other methods, none of them take into account the leakage of driver's facial information. In the method proposed in this paper, we use the template fusion method to fuse the driver's face and finally output the simulated image of the driver's face through 3D simulation, which fundamentally avoids the problem of driver's facial privacy leakage.

There are indeed some problems with 3D convolution. For the existing problems, we have given corresponding solutions in the process of model construction and experimentation:

(1) From the perspective of data format: The flaw of the 3D network lies in the problem of memory occupation, which makes it impossible to use the entire 3D image layer as input and must be cropped and cut into a series of 3D patches as input. The cropped blocks will limit the maximum receptive field that the network can achieve, resulting in the loss of certain global information. If the target to be segmented is itself much larger than the cropped blocks, it is difficult for the network to learn the overall structural information of the target. For this problem, we adopt the scheme of hardwired layer to introduce the information of five channels of grayscale, optical flow and gradient into the network, which encodes the prior knowledge of feature information and increases the richness of input information to reduce the impact of this information loss.

(2) From the perspective of model training: Generally speaking, the number of parameters of 3D convolution is larger, so the 3D convolution we commonly use are not like 2D convolution. The 3D convolution is not down sampled at a high multiplicity, but at a lower multiplicity. However, due to the matching problem between the amount of data and the amount of model parameters, the 3D convolutional neural network may require more data to train, otherwise it may lead to overfitting. In response to this problem, our approach is to expand the dataset as much as possible. As shown in Figure 7, we have collected a large number of fatigue detection datasets and in addition we have collected and produced our own datasets under real vehicle conditions and trained on a large amount of data to improve the generalization performance of the model. If there are enough training samples and the computing power is strong enough, the model can quickly fit to the optimal encoding function.

(3) From the perspective of problem context: Most of the deep learning-based driving fatigue detection methods process driver facial signals and the process of fatigue is usually a time-series problem. 2D networks only calculate features from the spatial dimension and the information of temporal features is often ignored, which can be improved by introducing the time-series module (LTSM, RNN) [37,38] to solve this problem, but this makes the complexity of the model higher and is not conducive to later real-world applications. 3D networks are able to capture inter-frame motion information, which is a good alternative solution for fatigue feature capture.

The channel attention mechanism module is introduced to allow the model to learn to focus on the important information and ignore the unimportant ones, thus allowing the extracted features to be more directional. For the fatigue detection task, the SE module is used to relearn the eye and mouth opening and closing features to obtain more accurate fatigue detection results. In the 'Squeeze' operation, the features are compressed in the spatial dimension and the feature channels are transformed into a real number, which has a global field of perception to some extent. The 'Excitation' operation then feeds the feature map into two fully connected networks to generate the weights of each feature channel; this processing of local information is essentially a hierarchy of information importance. The attention mechanism module simulates the human brain's response when seeing an image, first acquiring global information about the whole image to get a global overview of the whole image and then 'focusing' on the areas of interest to humans. Humans also acquire image information from the global to the local level, so the proposed scheme is logically successful.

## 6. Conclusions and Future Work

This study proposes a fatigue driving detection system based on the combination of 3D convolution and attention mechanism. First, we locate the key areas of the driver's face based on the Dlib library, then use 3D-SE-Net to extract the features of the eyes and mouth and finally output the fatigue state through a full binary tree SVM classifier. The experimental results show that the model can accurately extract the features of eyes and mouth and achieve satisfactory accuracy on the four datasets. At the same time, we have done a lot of control experiments on the attention mechanism module. The results show that the addition of the attention mechanism module can optimize the feature weight and improve the detection accuracy. We also use Grad-CAM to visualize the influence of different lighting conditions and whether wearing glasses on the model detection accuracy and the results show that 3D-SE-Net can still achieve better detection results in dark environments. On the YawDD dataset, the detection accuracy of the proposed model reaches a satisfactory 99.03%, which is more accurate and robust than other methods. Last but not least, we use a template fusion approach to design a facial privacy protection software that can record changes in the driver's facial status in real time while keeping the driver's privacy safe.

Our proposed method still has some shortcomings, the model is complex and the 3D convolution operation consumes a lot of computing power. Therefore, in the future, we consider using a residual network to simplify our model. In addition, we will expand our detection range to add more perspective driver behavior datasets to take into account the detection of driver distracted behavior in the car.

**Author Contributions:** Conceptualization, W.X. and W.Z.; methodology, W.X.; software, W.X.; validation, W.X. and C.L.; formal analysis, W.X.; investigation, W.X.; resources, W.X.; data curation, F.L.; writing—original draft preparation, W.X.; writing—review and editing, W.X.; project administration, W.Z.; funding acquisition, X.W. All authors have read and agreed to the published version of the manuscript.

**Funding:** This work was supported in part by the National Natural Science Foundation of China (No. 51805312).

**Institutional Review Board Statement:** Not applicable.

**Informed Consent Statement:** Not applicable.

**Data Availability Statement:** Not applicable.

**Acknowledgments:** The authors would like to express their appreciation to the developers of Pytorch and OpenCV, the authors of Grad-CAM and dataset provider School of Information Science, Beijing Language and Culture University.

**Conflicts of Interest:** The authors declare that they have no conflict of interest.

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
