# Peer review of "Driving Fatigue Detection Based on the Combination of Multi-Branch 3D-CNN and Attention Mechanism"

_applsci, doi:10.3390/app12094689_

Round 1

Reviewer 1 Report

This is a derivative type work in this domain. While some derivative works do provide new directions, in this case, the novelty is quite low and the contribution is not sufficient. The authors have tried to show gains on multiple aspects however, the overall gain is not decisive. In various cases, other schemes do better. Again, the scheme presented must be very computation intensive. The performance analysis shown is not really convincing and I suspect the appropriate measurements are not taken but rather selected scores are chosen. While the authors mention about taking cases of glass and with glass or so, the case of tinted glass is not well addressed or is not clear. In any case, I have the understanding that this work can be rather tried for a conference. A journal paper needs enough depth of technical content and new insights or novelty. The paper fails on these fronts. I would recommend rejection. I would not buy this idea.

Author Response

Dear Reviewers, We sincerely thank you for your comments on the revisions to our manuscript. We have revised the manuscript in accordance with your comments, as described in Word.

Reviewer 2 Report

I went through the paper and think it's an interesting topic.

  1. Some grammar mistakes should be concerned. for example, Its source code is implemented in C++ and provides a 180 Python interface. 
  2. The model obtains information of multiple channels of grayscale, gradient and optical flow from 18 the input frame, and then uses 3D convolution to extract features from spatial and temporal dimen-19 sions, and then feeds the feature map to the attention mechanism module to optimize feature 20 weights. A passive tense may be better in my opinion.
  3. The experimental results show that, compared with other classic fatigue 24 driving detection methods, this method extracts features from the temporal and spatial dimensions 25 and optimises the feature weights using an attention mechanism module, achieving better detection 26 performance, which can be effectively applied to driving fatigue detection. This sentence may be too long.
  4. The article structure is good and the figures are clear. 

Author Response

(The authors gave the same response as above.)

Reviewer 3 Report

This paper provides a system for detecting driving fatigue based on a combination of multibranch 3D-CNN and attention processes. The authors analyze fatigue criteria and classify driving states. However, the paper is well-organized and well-written.
Minor concerns should be considered on paper:
1. The author needs to mention the main scientific or methodological approach in the abstract. Also, the author has to write a complete description of EAR and MAR, when using it for the first time.
2. The most recent articles should be considered as a reference as well as comparing the original work.

Author Response

(The authors gave the same response as above.)

Round 2

Reviewer 1 Report

The revised version indeed looks better than the previous one. Rather than the technical contribution in the paper, I am appreciating basically the efforts given by the authors to revise the paper. With the additions of extra modules, it may have some extra features but fundamentally, the work has not changed. Still, this could be considered an average contribution paper and may be accepted with this version. My comments however, must be considered alongside the comments from the other reviewers.

Reviewer 2 Report

The questions I concerned are addressed and I think this paper can be considered for accepting once other reviewers' concerns were also addressed.